# Improvements in Sugarcane (*Saccharum* spp.) Varieties and Parent Traceability Analysis in Yunnan, China

Yong Zhao , Fenggang Zan, Jun Deng, Peifang Zhao, Jun Zhao, Caiwen Wu, Jiayong Liu * and Yuebin Zhang *

Sugarcane Research Institute, Yunnan Academy of Agricultural Sciences, Kaiyuan 661699, China;
18087395132@163.com (Y.Z.); fengang88@126.com (F.Z.); dj@yaas.org.cn (J.D.); hnzpf@163.com (P.Z.);
junzhao_ky@126.com (J.Z.); gksky_wcw@163.com (C.W.)
* Correspondence: lljjyy1976@163.com (J.L.); ynzyb@sohu.com (Y.Z.)

**Abstract:** Sugarcane (*Saccharum* spp.) breeding in China has a history of nearly 70 years. Yunnan province represents the second largest sugarcane planting area in China; therefore, by studying the evolution of sugarcane varieties in this region, it is possible to gain an understanding of the process of improvement since the foundation of sugarcane hybrid breeding. In this study, we compared the main industrial and agronomical characteristics of 107 sugarcane varieties, developed between 1952 and 2020, and discussed the reasons for replacement and exchange. Overall, significant differences were observed ($p < 0.01$), highlighting notable improvements, especially in terms of yield; however, disease incidence remains a serious issue and the fundamental reason for variety replacement. Meanwhile, analysis of parent traceability revealed that the main varieties cultivated at present have a similar parental relationship based around CP, F, and YC series germplasms. Taken together, these findings suggest that disease-resistant breeding should be strengthened, and susceptible varieties eliminated, while making full use of existing varieties as core parents.

**Keywords:** sugarcane (*Saccharum* spp.); variety improvements; agronomic/industrial characteristics; disease incidence; parent traceability



## 1. Introduction

Sugarcane (*Saccharum* spp.) is a gramineous C$_4$ plant and an important raw material in global sugar production. In 2020, the total sugarcane planting area in China was approximately 1.35 million hectares, with approximately 0.8 and 0.3 million hectares in Guangxi and Yunnan Provinces, respectively. Despite being the second largest sugarcane planting region in China, Yunnan, which is a mainly mountainous region, is experiencing a gradual decrease in planting area due to changes in the industrial structure and market supply, as well as planting competition with other crops. Improvements to modern sugarcane varieties and a fundamental breakthrough in sugarcane yield and sucrose content are required in order to protect the interests of sugarcane growers and sugar industries nationwide.

Roach (1989) delineated three phases in modern (since about 1890) sugarcane improvement as follows. The first phase focused on crossing and selecting among *S. officinarum* clones. Most early commercial sugar industries around the world were based on several clones of this species. The second phase involved development of interspecific hybrids between *S. officinarum* and other species, principally *S. spontaneum*. The third phase involved exploitation of the interspecific hybrids developed in phase two [1]. The cultivation history of sugarcane in China extends more than 2000 years, but sugarcane hybrid breeding was established in 1952 by the Hainan Sugarcane Breeding Factory [2], and since then, more than 300 varieties have been developed and approved [3,4]. With resulting improvements in sugarcane characteristics, many new varieties have been introduced to Yunnan and the core varieties are continuously updated. While some varieties such as ROC22 have dominated for decades [5], others such as Yunzhe (YZ) 09-1601 have been eliminated due

to the process of popularization, or as a result of national regional testing. Moreover, other varieties once preferred by sugarcane growers and sugar industries such as Yuetang (YT) 93159 have since been phased out. Analysis of the evolution of sugarcane varieties since the establishment of hybrid breeding programs and the reasons for elimination or extinction may help guide the future direction of crossbreeding. At the same time, parent traceability analysis in dominant sugarcane planting regions and an understanding of the role of traceable parents in modern sugarcane improvements may also provide an important reference for future utilization.

Hybrid crossbreeding forms the basis of improvements in sugarcane varieties [6]. Development of sugarcane planting in Yunnan has undergone five generations of improvement. The first was represented by *Saccharum*, the second by F134, the third by Guitang (GT) 11 and YZ71388, the fourth by ROC22 and YT93159, and the fifth by Liucheng (LC) 05136, YZ081609, and YZ0551. As a result, the sucrose content and overall yield of sugarcane in Yunnan Province are now ranked highest nationwide [7]. However, according to current research, the gene bank for sugarcane breeding is poor, the genetic relationships between sugarcane varieties are very similar, and the genetic basis is narrow [8]. Therefore, achieving notable breakthroughs in sugarcane crossbreeding is difficult, a problem faced by sugarcane breeding programs around the world [9–11].

In addition to the creation of new core parents, it is important to make full use of existing varieties in order to achieve breeding breakthroughs [12]. For example, in the United States, the use of recurrent selection technology has effectively realized improvements in sugarcane varieties [13]. At present, the main goal of sugarcane crossbreeding is to improve and enhance agronomic and industrial characteristics, notably yield characteristics and sucrose content. In the past 50 years, world sugarcane production has increased almost three-fold, with production gains partly attributed to genetic improvements adapted to particular target environments [14].

The replacement of existing varieties is usually due to the introduction of new and improved material, with the success of crossbreeding based on the scientific selection of cross-parent combinations. In general, high-sugar parents have more stable heritability, and are easier to implement in the cultivation of high-sugar hybrid offspring [15]. At present, China has a large collection of sugarcane resources, the evaluation of which is largely based on quantitative industrial and agricultural characteristics [16,17]. However, such evaluation is time consuming; moreover, the findings are often irrelevant. In contrast, evaluation based on data grading is widely used in germplasm analysis and breeding, with good results having been achieved. For example, in sugarcane breeding, the division of general vigor into nine levels revealed that the resulting genetic variation was greater than that of sugarcane brix in the first stage of breeding [18]. Similarly, we previously evaluated 317 sugarcane germplasms based on classification of agronomic characteristics, saving a lot of field time and improving the overall evaluation efficiency [19].

Establishment of the sugarcane hybrid parent system in China occurred as a result of a combination of introduced foreign germplasms and independent innovation. Foreign germplasm resources have played an important role in the cultivation of new sugarcane varieties in China, and according to incomplete statistics, from 1950 to 2000, 163 varieties were bred from 21 parents including CP, F, and Co series, which account for 87.63% of the counted varieties, showing the high inbreeding coefficient [3]. Therefore, CP, F, and Co series germplasms have become true core parents, while a previous report into the utilization of sugarcane germplasm resources in mainland China found that the CP and Co germplasms play an important role in evolving improvements [20]. Researchers also suggested that the commercial success of Co205 gave new direction to sugarcane breeding programs in almost all sugarcane growing countries [21].

In recent years, China has bred a large number of new sugarcane varieties using these core germplasms. These varieties not only occupy a certain position in production, but have also resulted in new core germplasms, playing an important role in further improvements. The aim of this study is to compare the sugarcane varieties or materials that have

been planted, popularized, or cultivated in Yunnan Province since the development of sugarcane hybrid crossbreeding in China. The main industrial and agricultural characteristics, including biomass yield and sucrose scores, were then compared. The aim was to explore the reasons why certain sugarcane varieties have become either dominant or eliminated, thereby providing a foundation for future breeding programs. In addition, through parent traceability analysis, this paper also studied the role of traceable parents in modern sugarcane improvements, highlighting potential parent references for future cross utilization.

## 2. Materials and Methods

### 2.1. Experimental Materials

A total of 107 sugarcane varieties with a history of planting or current cultivation in Yunnan Province were included. Of these, 29 belonged to the "Yunzhe" series, 3 to the "Chuantang" series, 3 to the "Dezhe" series, 4 to the "Funong" series, 15 to the "Ganzhe" series, 2 to the "Huanan" series, 4 to the "Liucheng" series, 4 to the "Mintang" series, 12 to the "Yuetang" series, 16 to the "Guitang" series, and 15 to the "ROC" series, imported from Taiwan (China) (Table 1).

**Table 1.** Test varieties and breeding institutions where they were developed.

| Series * | Breeding Institution | No. of Varieties |
|---|---|---|
| YZ | Sugarcane Research Institute of Yunnan Academy of Agricultural Sciences | 29 |
| CT | Plant Engineering Research Institute of Sichuan Province | 3 |
| DZ | Sugarcane Research Institute of Dehong Prefecture, Yunnan Province | 3 |
| FN | Fujian Agriculture, and Forestry University | 4 |
| GZ, GN | Gannan Academy of Sciences | 15 |
| HN | Former South China Institute of Agricultural Sciences | 2 |
| LC | Guangxi Liucheng Institute of Agricultural Sciences | 4 |
| MT | Fujian Academy of Agricultural Sciences | 4 |
| YT, YG | Biological Engineering Institute of Guangdong Academy of Sciences | 12 |
| ROC | Materials imported from Taiwan, China | 15 |
| GT | Sugarcane Research Institute of Guangxi Academy of Agricultural Sciences | 16 |

* YZ: "Yunzhe", CT: "Chuantang", DZ: "Dezhe", FN: "Funong", GZ: "Ganzhe", GN: "Gannan", HN: "Huanan", LC: "Liucheng", MT: "Mintang", YT: "Yuetang", YG: "Yuegan", ROC: "Xintaitang", GT: "Guitang".

### 2.2. Experimental Site and Test Design

Some researchers have documented that the main sugarcane-producing areas in Yunnan, such as Baoshao, Dehong, Lincang, and Kaiyuan, possess similar environmental and climatic characteristics, and the G×E interaction showed no significance [22]. Kaiyuan, the main sugarcane breeding region in Yunnan, where the local climate is characterized by high humidity and temperature that would contribute to the high rates of diseases, could meet our demands for professionals and special equipment, which were needed for the detection of the industrial characteristics. A field experiment was carried out in the first scientific research base of the Sugarcane Research Institute, Yunnan Academy of Agricultural Sciences (Kaiyuan city, Yunnan Province; 23.7° N, 103.25° E), on 20 December 2016. The experiment adopted a randomized complete block design, with two repetitions. Each variety was planted in 10 rows, each 8.0 m in length and spaced 1.1 m apart. The experimental plot included irrigation systems, and management was consistent with field production. Agronomic and industrial characteristics were investigated at maturity, about December in the next 3 years.

Agronomic characteristics were investigated based on a previous report using the data grading method [18]. The range of agronomic characteristics was determined based on the grading requirements of this experiment and breeders' experience. The survey was divided into three components: new plant (2017), ratoon 1 (2018), and ratoon 2 (2019), and the survey time was the beginning of December in each year. First, a team composed of 3–4 sugarcane breeders (each characteristic was investigated by a specially assigned person,

except the general vigor; the team then discussed grading together and members remained unchanged throughout grading) graded the 107 varieties for the following agronomic characteristics: plant height, stem diameter, millable stalk count, the incidence of leaf diseases (field diseases such as mosaic disease, brown rust, smut, brown streak, and spot disease), and general vigor. Each characteristic was then graded from one to five with reference to Zhao et al. [18]. Details were as follows: height from high (grade 1) to low (5); stem diameter from thick (1) to thin (5); millable stalk count from maximum (1) to minimum (5); leaf disease incidence from absent or mild (1) to severe (5); and general vigor from vigorous (1) to weak (5), as shown in Table 2. The classification of disease incidence was determined according to the presence of mosaic disease, brown rust, smut, and spot disease. The weight of single sugarcane stems was measured directly. Six stems were randomly selected per variety (three per replication) and average values were calculated.

**Table 2.** Grading of each agronomic characteristic.

| Rank * | Height (cm) | Stem Diameter (cm) | Millable Stalk Count (m$^2$) | Incidence of Leaf Disease (%) | General Vigor |
|---|---|---|---|---|---|
| 1 | >280 | >3.00 | >10 | <5 | From vigorous (1) to |
| 2 | 220~280 | 2.5~3.00 | 8~10 | 5~10 | weak (5) based on |
| 3 | 160~220 | 2.00~2.50 | 6~8 | 10~20 | comprehensive |
| 4 | 100~160 | 1.50~2.00 | 4~5 | 20~30 | observations of |
| 5 | <100 | <1.50 | <4 | >30 | overall performance. |

* Grading was based on the evaluation standard of sugarcane agronomic characteristics in China and the long-term selection experience of the breeders.

As industrial characteristics, sugarcane brix, sucrose content, juice sucrose content, fiber content, and gravity purity were determined with reference to a previous report [23]. Juice was extracted from three stalks, selected randomly from each plot, using a mechanical cane juicer in all environments. The extracted juice was analyzed for brix (%) using an automatic refractometer, Rudolph J257 (Rudolph Research Analytical, USA), and the sucrose content (%), juice sucrose content (%), and gravity purity (%) were measured using an automatic saccharimeter, Autopol 880 (Rudolph Research Analytical, USA). After extracting juice, the remaining cane residue was weighed and oven-dried to determine the fiber content (%). New plantings were sampled within the middle ten days of each month from November 2017 to March 2018, ratoon 1 was sampled monthly from November 2018 to March 2019, and ratoon 2 was tested monthly from November 2019 to March 2020.

### 2.3. Experimental Climate Conditions

Yunnan Province has a subtropical plateau monsoon-type climate, with remarkable three-dimensional climate characteristics, distinct dry and wet seasons, and abnormal changes in temperature with vertical terrain. The Sugarcane Research Institute of Yunnan Academy of Agricultural Sciences is located at an altitude of 1051.8 m and has sufficient light resources. In 2017, the annual sunshine duration was 1960 h, with an annual average temperature of 20.1 °C, and a maximum and minimum of 34.1 °C and 3.3 °C, respectively. Annual precipitation was 1038.4 mm and annual potential evaporation was 1987 mm. The frost-free period was approximately 341 days. In 2018, the sunshine duration was 2125.3 h, the annual average temperature was 21.5 °C (maximum: 37.7 °C, minimum: 0.2 °C), annual precipitation was 698.2 mm, annual potential evaporation was 1880 mm, and the frost-free period was around 320 days. In 2019, the sunshine duration was 2033.7 h, the annual average temperature was 20.8 °C (maximum: 36.2 °C, minimum: 4.2 °C), annual precipitation was 592.0 mm, annual potential evaporation was 1860 mm, and the frost-free period was around 330 days.

### 2.4. Data Analysis

Microsoft Excel 2019 was used for data sorting and histogram of production area making. Multiple comparisons were analyzed by an LSD (post hoc) test by the software DPS v14.10 (Zhejiang University, Hangzhou, China). R software was used for figure

drawing and the Kruskal–Wallis test and LSD for the difference analysis of each cluster-variety group. The professional online mapping tool, Process On (Beijing Barley Land Information Technology Co., Ltd. China), was used to create the parent traceability chart.

## 3. Results

### 3.1. Variation in Agronomic and Technological Characteristics among the 107 Varieties

Analysis of variance of the agronomic characteristics showed significant differences, including single-stem weight, sugarcane height, stem diameter, millable stalk count, leaf disease incidence, and general vigor between the varieties for the new plants, ratoon 1, and ratoon 2 ($p < 0.01$, coefficient of variation: 20.21–38.23%; Table 3).

**Table 3.** Variance analysis of the agronomic characteristics in 107 sugarcane varieties.

| Characteristic | Newly Planted | | Ratoon 1 | | Ratoon 2 | |
|---|---|---|---|---|---|---|
| | Mean | CV (%) | Mean | CV (%) | Mean | CV (%) |
| Single-stem weight (kg) | 1.46 ** | 20.21 | 1.20 ** | 23.97 | 0.83 ** | 24.01 |
| Height | 2.28 ** | 36.47 | 2.66 ** | 38.82 | 2.37 ** | 35.36 |
| Stem diameter | 2.44 ** | 36.43 | 2.53 ** | 39.51 | 2.46 ** | 34.89 |
| Millable stalk count | 2.88 ** | 30.56 | 2.93 ** | 36.20 | 2.88 ** | 28.92 |
| Leaf disease incidence | 2.74 ** | 36.79 | 2.80 ** | 43.15 | 2.71 ** | 33.73 |
| General vigor | 2.66 ** | 38.23 | 2.79 ** | 37.74 | 2.58 ** | 36.74 |

** $p < 0.01$ according to the LSD test. Except for single-stem weight, all characteristics were graded using the system shown in Table 2.

Similarly, analysis of variance of the industrial characteristics revealed significant differences in sugarcane brix, juice sucrose, sugarcane sucrose, gravity purity, and sugarcane fiber ($p < 0.01$, coefficient of variation: 6.05~13.04%; Table 4).

**Table 4.** Variance analysis of the industrial characteristics.

| Characteristic | Newly Planted (%) | | Ratoon 1 (%) | | Ratoon 2 (%) | |
|---|---|---|---|---|---|---|
| | Mean | CV | Mean | CV = | Mean | CV |
| Sugarcane brix | 21.34 ** | 6.92 | 21.75 ** | 5.93 | 22.47 ** | 6.05 |
| Juice sugar content | 18.41 ** | 9.43 | 18.82 ** | 7.69 | 19.64 ** | 7.95 |
| Cane sugar content | 14.78 ** | 9.21 | 15.00 ** | 7.18 | 15.65 ** | 7.16 |
| Gravity purity content | 84.63 ** | 4.82 | 86.27 ** | 3.53 | 87.31 ** | 3.47 |
| Cane fiber content | 13.38 ** | 13.04 | 15.09 ** | 12.13 | 15.26 ** | 12.53 |

** $p < 0.01$ according to the LSD test.

### 3.2. Cluster Analysis

3.2.1. Cluster Analysis Based on the Agronomic and Industrial Characteristics

Cluster analysis grouped the 107 sugarcane varieties into four groups (Figure 1). Group 1 included 11 varieties, group 2 included 16, group 3 included 36, and group 4 included 44.

3.2.2. Differences in Agronomic and Industrial Characteristics among Cluster Groups

Analysis of agronomic characteristics among the four cluster groups revealed significant differences in the single-stem weight (Figure 2A; $p < 0.001$). There were also significant differences in sugarcane height between all groups, except groups 1 and 2 ($p < 0.001$; Figure 2B), and in leaf disease incidence, except between groups 1 and 3 ($p < 0.01$; Figure 2C). Significant differences in stem diameter were observed among all 4 groups ($p < 0.001$; Figure 2D), and in the millable stalk count between all groups, except groups 1 and 3 ($p < 0.001$; Figure 2E). Meanwhile, there were extremely significant differences between groups in terms of general vigor ($p < 0.01$; Figure 2F).

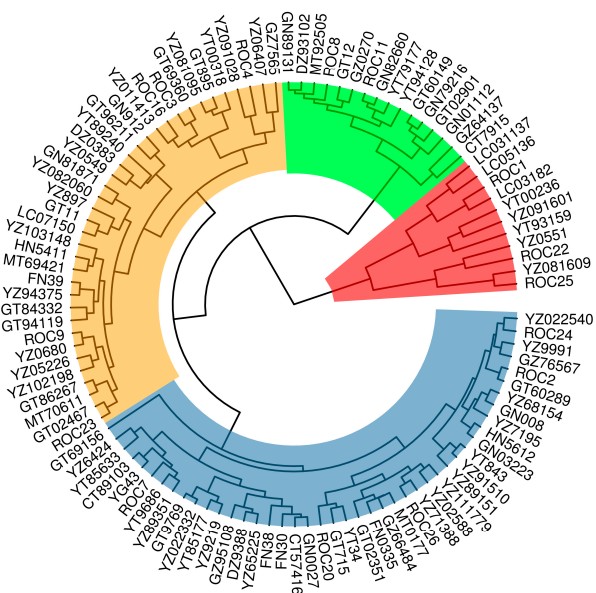

**Figure 1.** Cluster analysis of the 107 sugarcane variety resources examined here. The red area represents group 1, green represents group 2, yellow represents group 3, and grey represents group 4. Agronomic characteristic data were processed using a reciprocal approach, while the industrial characteristics were processed via an arcane process, then all data were normalized.

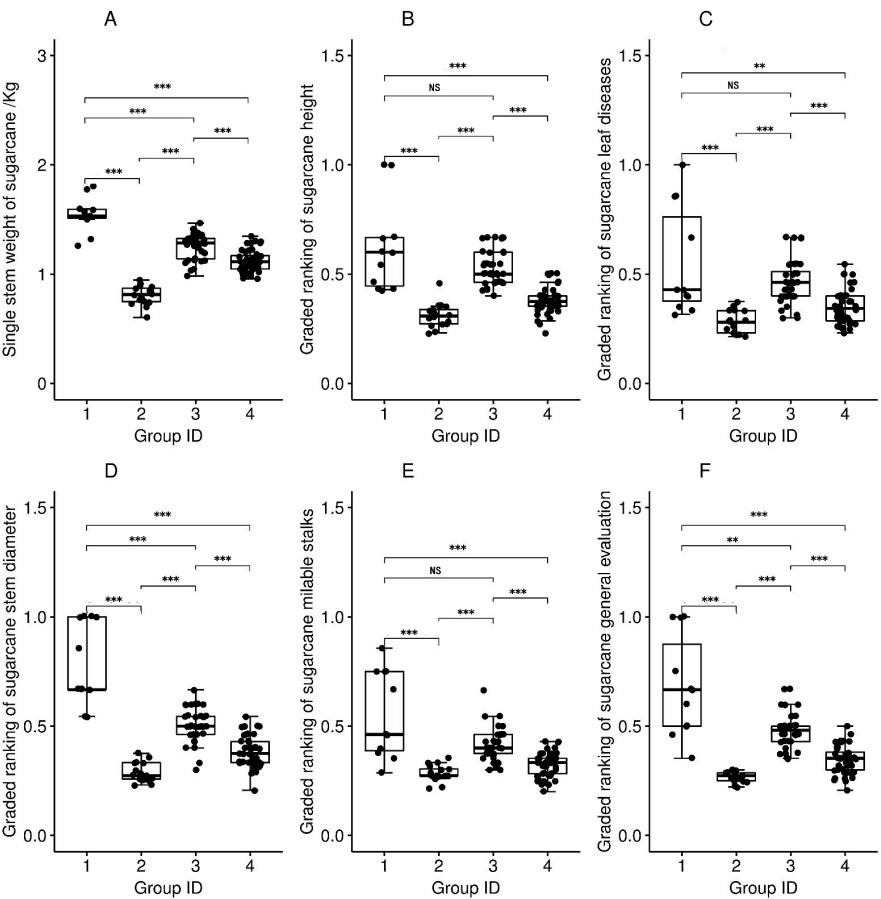

**Figure 2.** Differences in agronomic characteristics between the four cluster groups. Comparisons of (**A**) single-stem weight, (**B**) height, (**C**) leaf disease incidence, (**D**) stem diameter, (**E**) millable stalk count, and (**F**) general vigor are shown based on grading data of each cluster-variety group. Not significant (NS), $p > 0.05$; **, $p < 0.01$; ***, $p < 0.001$ by LSD.

In contrast, analysis of industrial characteristics among the four groups revealed no significant differences in terms of brix, juice sucrose, cane sucrose, fiber, and gravity purity ($p > 0.05$; Figure 3).

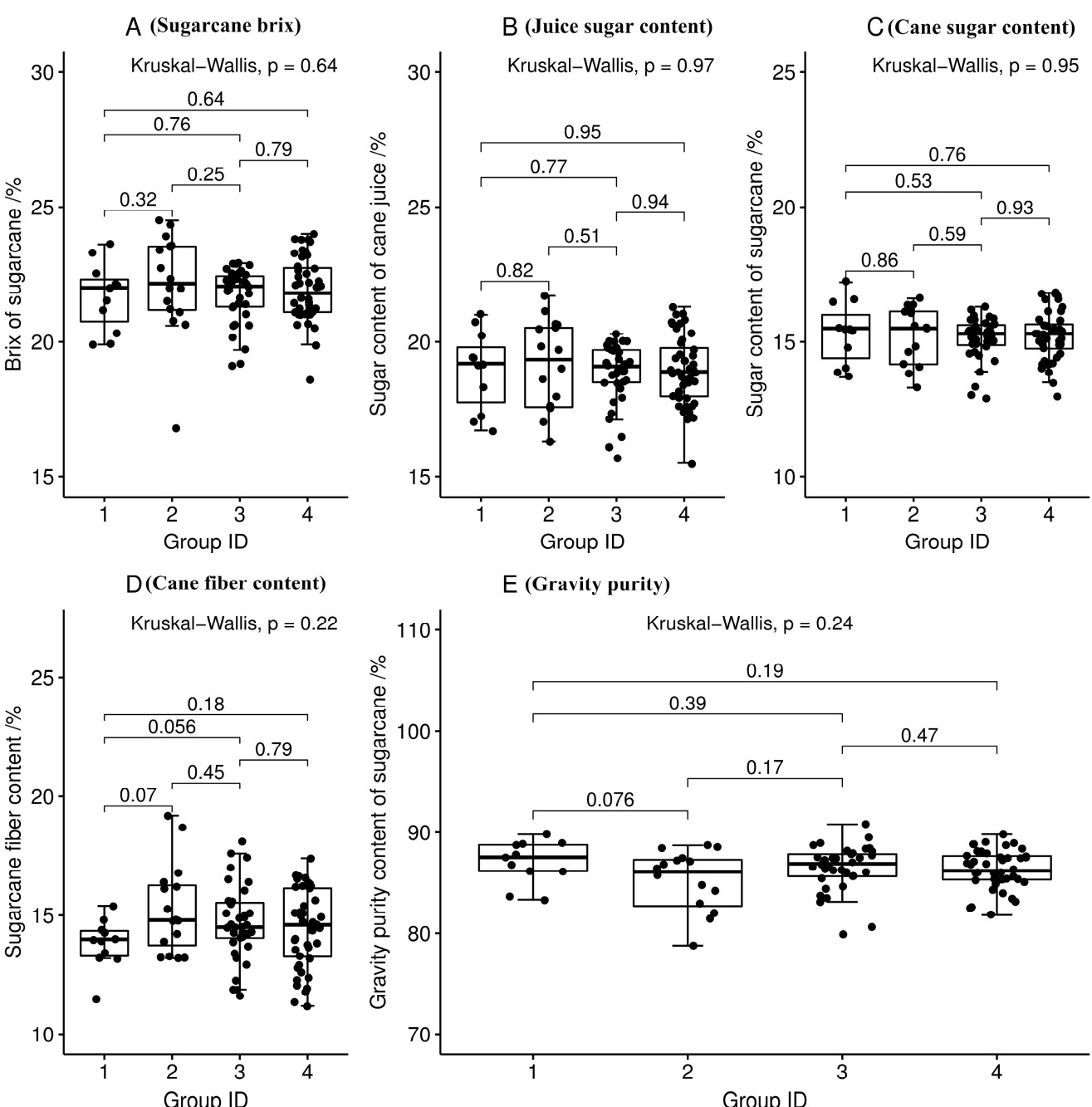

**Figure 3.** Differences in industrial characteristics between the four cluster groups. Comparisons of (**A**) sugarcane brix, (**B**) juice sugar content, (**C**) cane sugar content, (**D**) cane fiber content, and (**E**) gravity purity are shown. R statistics Kruskal–Wallis test was used for the difference analysis.

### 3.2.3. Differences in the Cultivated Area of the four Cluster Groups

In 2020, the sugarcane planting area in Yunnan represented approximately 2.73 million hectares. Of this, group 1 represented approximately 2.09 million hectares, with all other clusters combined representing around 0.64 million hectares. The 11 sugarcane varieties in group 1 accounted for about 76.63% of the total cultivated area in Yunnan. Among these, ROC22 represented around 45.3 thousand hectares, YT93159 around 44.5 thousand

hectares, YZ0551 around 24.7 thousand hectares, LC05136 around 24.1 thousand hectares, YZ081609 around 12.67 thousand hectares, ROC1 around 19.3 thousand hectares, ROC25 around 17.87 thousand hectares, LC031137 around 7.27 thousand hectares, YT00236 around 9.46 thousand hectares, LC03182 around 3.7 thousand hectares, and YZ091601 around 133.3 hectares (Figure 4).

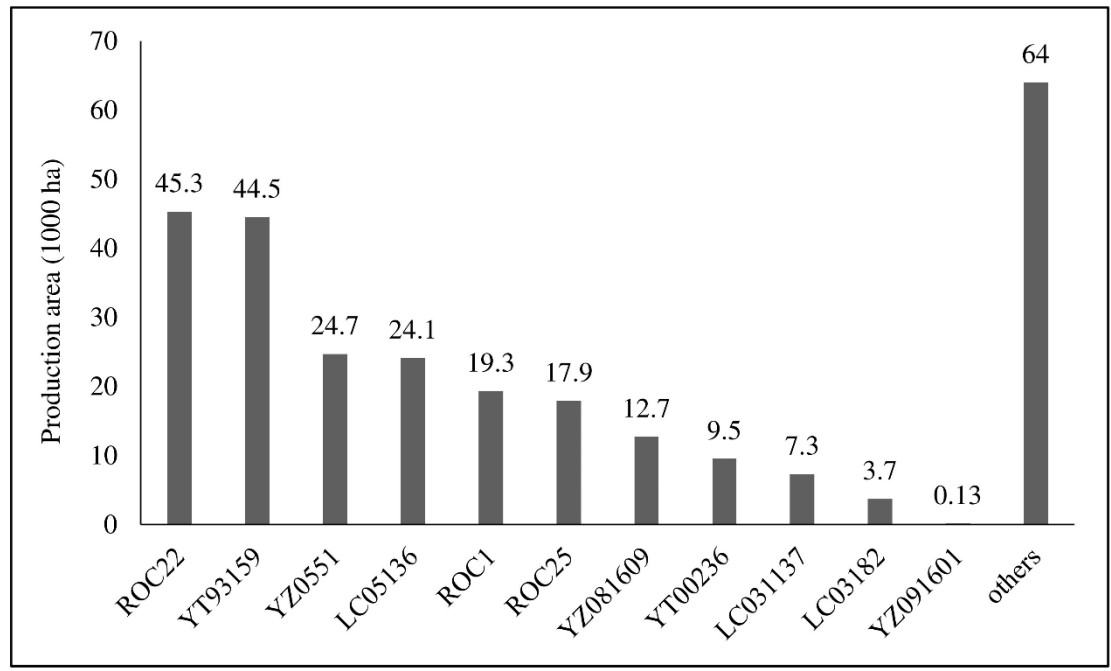

**Figure 4.** Area of sugarcane planting in Yunnan Province in 2020 by variety.

### 3.3. Parent Traceability Analysis of the Main Cultivated Varieties (Group 1)

Parent traceability analysis of the sugarcane varieties in group 1 subsequently revealed that nearly all had an F or CP background (Figure 5). CP721210 was identified as an important parent of a number of Yunnan sugarcane varieties, while F134, F108, and F146 are the core parents of early sugarcane breeding in China. ROC22 not only plays a significant role in sugarcane production, but was also the male parent of LC03182, LC05136, and LC031137, all of which have an important role in breeding improvements. Similarly, parents of the YC series germplasm, such as YC8545 and YC 8296, played an important role in improving varieties YZ0551 and YZ081609, both of which were bred under the Yunnan *Saccharum spontaneum* background. The parents of ROC25 are 79-6048 and 69-463; however, information on these two varieties is currently lacking, so their background relationships could not be determined. YN73204 was identified as an important female parent of the YT series.

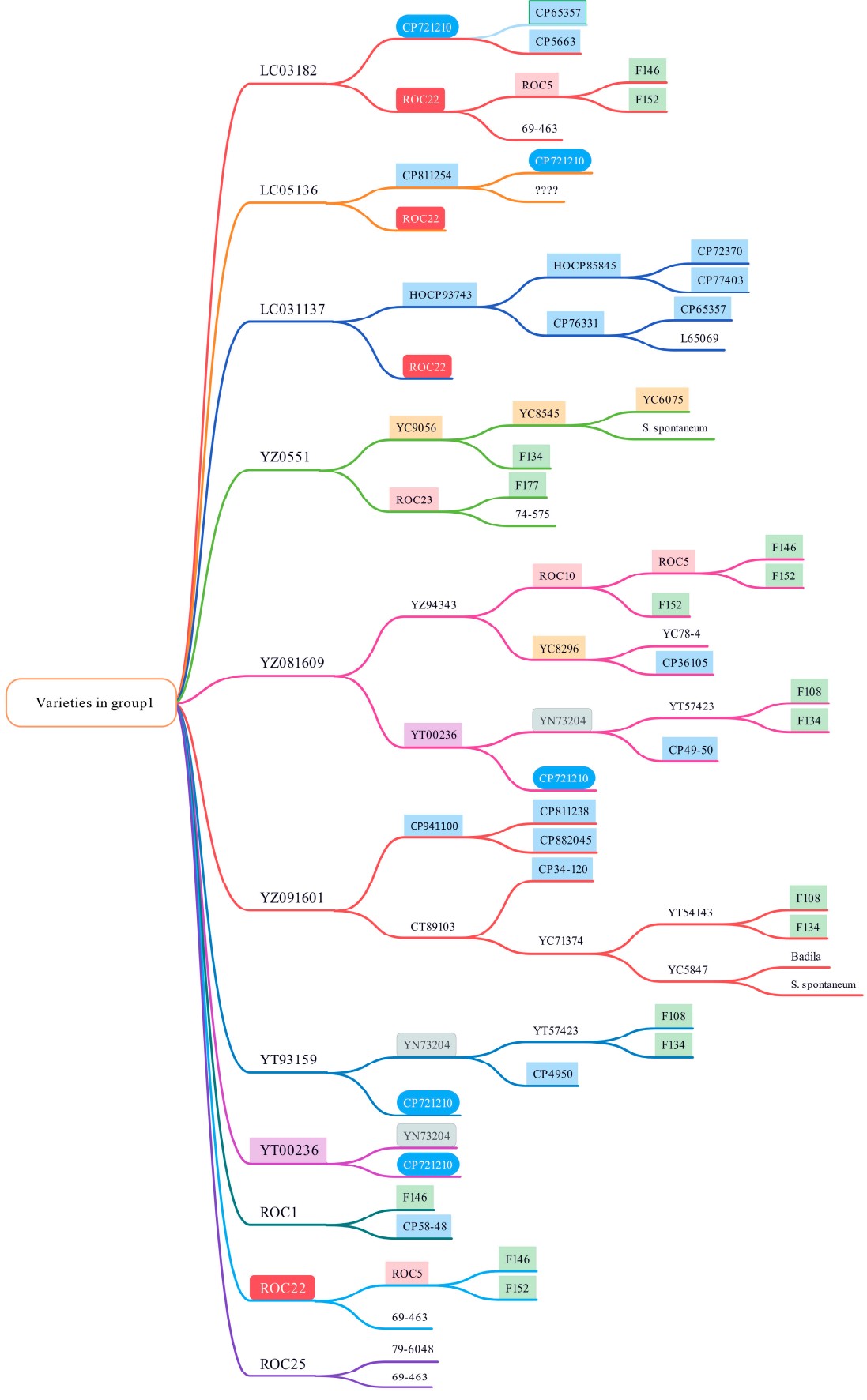

**Figure 5.** Parent traceability analysis of the group 1 varieties. "?": no parental information available.

## 4. Discussion

### 4.1. Improvements in Sugarcane Varieties in Yunnan

Since the development of sugarcane hybrid crossbreeding, many varieties have been developed, thereby replacing previous varieties cultivated in Yunnan sugarcane regions. This study showed significant differences in the industrial and agronomic characteristics of 107 varieties. The coefficients of variation also varied widely from 20.21–38.23% and 3.47–22.47% in terms of the agronomic and industrial characteristics, respectively, suggesting that the quality of sugarcane varieties has changed during development. Comparisons and statistical analyses of the resulting cluster groups indicated that the varieties in group 1 have a significantly greater height, stem diameter, millable stalk count, single-stem weight, and general vigor than all other groups. Because these varieties represent the majority of cultivated sugarcane in Yunnan at present, it can be concluded that the overall yield has also experienced a significant advance as a result of these improvements. This is also consistent with both nationwide and global trends. For example, analysis of 10 GT series varieties bred in China in the past five years showed an increase in yield of 0.54–29.49% compared to the control, with an advance in sucrose content of 0.33–1.33% [24]. Researchers suggested that the overall advance in sugarcane production in Brazil has greatly benefited from the development of new varieties, which have contributed to a continuous advance in yield [25]. Moreover, in Australia, new varieties are being bred based on advances in the productivity, sustainability, and competitiveness of Australia's sugarcane industry [26].

However, despite these findings, this study also revealed that the structure of varieties cultivated in Yunnan remains unreasonable. For example, older sugarcane varieties, such as ROC1, ROC22, and ROC25, still account for the majority of cultivated sugarcane, with newly bred varieties, such as YZ0551, YZ081609, and LC05136, representing a low proportion. No significant differences in sucrose content or other quality characteristics were observed between group 1 and the remaining three groups, which suggests that the main cultivated varieties do not play a significant role in developing the sugarcane industry; therefore, use of new varieties should be encouraged. At present, despite global improvements in the sucrose content and other quality characteristics, no major breakthroughs have been achieved, and issues remain. For example, in Florida, observed increases in sucrose yield in recently developed varieties were found to be associated with an increase in total cane yield, rather than improvements in commercial cane sugar [27]. Moreover, similar results were reported in three small-scale studies conducted in Australia, with no significant differences in commercial cane sugar between old and new varieties [28].

### 4.2. Field Diseases Remain the Biggest Threat to Sugarcane Production in Yunnan

The incidence rate of sugarcane leaf and field diseases is an important index of the quality of new varieties, as well as an important factor affecting their popularization and application [29]. Disease incidence directly affects the yield and sucrose quality, causing a reduction in both [30]. Disease is also a major problem facing sugarcane production worldwide [31], not only in China, and the results of this study suggest that the incidence rate remains high in both newly planted and ratoon crops. In this study, the average grading data of the 107 sugarcane varieties was around 2.75, which corresponds with the grading standard, while the incidence rate ranged from 10–20%. At present, the main sugarcane diseases in Yunnan Province are sugarcane smut [32], mosaic disease [33], brown rust disease [34], spot disease [29], and white leaf disease [35], occurrence of which seriously affects the popularization and application of new varieties. For example, YZ091601, which was developed by the Yunnan Sugarcane Research Institute, shows excellent performance in terms of yield and sucrose content; however, following wide-spread application and popularization, crops suffered large-scale brown rust disease. As a result, this variety is now cultivated in an area of approximately 133.3 hectares. Meanwhile, due to its wide adaptability and high yield, the ROC22 variety once occupied more than 70% of the total production area of mainland China following its introduction from Taiwan, making it the dominant variety. However, as a result of long-term planting, ratoon smut disease has become a

serious issue, and it has since experienced a gradual decrease in popularity; as a result, its production area now represents less than 30%, with the risk of complete elimination. Moreover, as a representative of new-generation sugarcane varieties in Yunnan, YZ081609 initially showed excellent performance in terms of sucrose content and yield, making it a top choice of sugarcane growers and sugar producers [36]. However, the occurrence of smut disease has since been widely observed with increasing severity. Therefore, breeding of disease resistance is urgently required, although progress remains slow both in China and worldwide. In contrast, sugarcane diseases are becoming increasingly diverse and serious [37].

### 4.3. CP, F, and YC Series Germplasms Play an Important Role in Improving Sugarcane Varieties in Yunnan

Analysis of sugarcane cultivation in Yunnan revealed that the varieties in group 1 account for approximately 76.63% of the total, highlighting their important role in production. We examined the parent traceability of group 1 and found that the background relationships were very similar, with many varieties sharing a common parent or relative. For example, seven main varieties (LC03182, LC05136, LC03-1137, YZ081609, ROC1, YT93-159, and YT00236) had a CP background, while all varieties in group 1 had an F series relationship, except ROC25, for which no parental information was available, and many CP series germplasms were also found to have an F background (Figure 5). Meanwhile, YZ0501 and YZ081609 have a YC background, which was created by hybridization of wild Chinese germplasms with F and CP series varieties, under the *S. spontaneum* background. The YC series germplasm plays an important role in sugarcane crossbreeding Yunnan, as well as representing core parents in sugarcane breeding nationwide. At present, more than 100 YC samples are used in sugarcane crossbreeding programs in China every year, accounting for 30% of all crossing parental lines [3]. Some new parents have been highly successful in breeding and utilization via introgression breeding with *S. spontaneum* and *E. arundinaceus*, such as YC71373, YC05164, and YC9747, resulting in the release of more than 30 commercial varieties [38], with the YC germplasm possessing the highest level of genetic diversity [39]. The genetic diversity of parental lines used in sugarcane breeding programs in China is continually increasing due to the introduction of new germplasms, such as CP72-1210 and ROC22, thereby supporting the future development of sugarcane breeding in China [40]. The CP germplasm also plays an important role in global sugarcane hybrid breeding [41,42].

### 4.4. Importance of Strengthening the Application of Main Existing Varieties as Core Hybrid Parents

The narrow genetic basis of Chinese sugarcane hybrid parents has been recognized. Breakthrough improvements should therefore focus on two main principles: the discovery and application of new wild resources and expansion of sugarcane kinship, and application of existing core parents or commercial varieties showing excellent performance [43]. The latter is the main approach currently being adopted by Chinese sugarcane breeders, because good parents are more likely to cultivate improved varieties, which is the basis of hybrid crossbreeding [44]. Under the premise that the new blood relationship of sugarcane has not been created yet, full use of high-quality commercial varieties for new cross utilization is one of the most effective methods of conventional breeding. In this study, parent traceability analysis revealed that some varieties represent both good commercial varieties and good hybrid parents (Figure 5). For example, ROC22, ROC23, and YT00236 not only play an important role in sugarcane production, but are also important hybrid parents. ROC22 is the male parent of LC03182, LC05136, and LC031137, while YT00236 is the female parent of YZ081609, and ROC23 is the female parent of YZ0551. In the broad sense, this phenomenon reflects recurrent selection, which plays an important role in sugarcane hybrid breeding systems in the United States and made an important contribution to sugarcane improvements. Moreover, recurrent selection for sucrose content was previously found to change growth and sugar accumulation in sugarcane [45]. Considering the sugarcane breeding programs

in Louisiana, USA, sucrose content is the top priority because the short growing season limits cane yield. To achieve this, a recurrent selection strategy is used to cross cultivars with a high sucrose content; then, a new generation of cultivars is selected from the progeny [13]. Effective improvements may be realized through recurrent selection technology based on existing rather than new parental relationships. Accordingly, current excellent varieties, such as YZ0551, YZ081609, and LC05136, may be used as excellent hybrid parents, allowing the realization of a new generation of improved sugarcane varieties.

## 5. Conclusions

Since the establishment of sugarcane crossbreeding in China, improvements have been made in the varieties cultivated in Yunnan Province, especially in terms of yield, which has experienced significant increases. After years of development, the structure of Yunnan sugarcane production is now represented by around 10 main varieties, all of which have similar CP, F, and YC series backgrounds. These germplasms also play important roles in sugarcane breeding nationwide. It is therefore important to make full use of these existing parental lineages, while strengthening the utilization of new-generation resources, such as YZ0551, YZ081609, and LC05136, in order to realize further improvements, especially in terms of disease resistance. The natural incidence of sugarcane disease in Yunnan remains serious, and that of sugarcane smut, brown rust disease, spot disease, and mosaic disease is high, which is the main reason why varieties are ultimately replaced. Therefore, the results of this study highlight the urgent need for disease-resistant breeding and an increase in the number of disease-resistant varieties.

**Author Contributions:** Conceptualization and design, Y.Z. (Yong Zhao); Writing-original draft, Y.Z. (Yong Zhao); Experiments, F.Z., J.D. and P.Z.; Data Analysis, J.Z. and C.W.; Project administration, Y.Z. (Yuebin Zhang) Writing—Review & Editing, J.L. and Y.Z. (Yuebin Zhang) All authors have read and agreed to the published version of the manuscript.

**Funding:** National Natural Science Foundation of China, Grant/Award Numbers: 32060505; 31660418; National sugar industry technology system, Grant/Award Numbers: CARS-170101; Yunnan provincial grants, Grant/Award Numbers: 202101AT070273; GDWG-2018-015; 202004AC100001-A02; 2019HC013; 2019IB008.

**Data Availability Statement:** Not applicable.

**Acknowledgments:** We thank Zhiyuan Wang and Panlei Wang for assistance with the data analysis and the use of the R software.

**Conflicts of Interest:** The authors declare no conflict of interest.

## Abbreviations

The following abbreviations are used in this manuscript:

| | |
|---|---|
| CT | Chuantang variety |
| DZ | Dezhe variety |
| FN | Funong variety |
| GN | Gannan variety |
| GT | Guitang variety |
| GZ | Ganzhe variety |
| HN | Huanan variety |
| LC | Liucheng variety |
| MT | Mintang variety |
| ROC | Taiwan variety |
| YG | Yuegan variety |
| YT | Yuetang variety |
| YZ | Yunzhe variety |

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
