# Peer review of "Improvements in Sugarcane (Saccharum spp.) Varieties and Parent Traceability Analysis in Yunnan, China"

_agronomy, doi:10.3390/agronomy12051211_

Round 1

Reviewer 1 Report

This manuscript reported the agronomic and industrial traits of sugarcane varieties released from 1952-2020 in Yunnan and parents of the varieties., and discussed the direction of sugarcane improvement in Yunnan. 

The author discussed the sugarcane production all over Yunnan, however, they evaluated only one environment and no replication of year in each crop type. Yunnan province is large area and there should be G x E in varieties. So, the author should be mentioned that why this experiment is enough for this discussion.

Furthermore, the author showed the parents of varieties but not show the inbreeding coefficient to evaluate genetic variation of the varieties. I think if the author show inbreeding coefficient should be better for the discussion.

Author Response

Responses to Reviewer 1 Comments

Comments 1:

  1. The author discussed the sugarcane production all over Yunnan, however, they evaluated only one environment and no replication of year in each crop type. Yunnan province is large area and there should be G×E in varieties. So, the author should be mentioned that why this experiment is enough for this discussion.

We appreciate the helpful suggestions. Here are our responses:

1.1 Characterized as C4 plant, sugarcane is a perennial crop. Usually, it can be harvested for 4 to 6 times or even more during the growing stage. In 1st year, we use sugarcane buds to generate the new plant and the stage calls New Plant. In 2nd year, sugarcane stems developed from the remained roots of 1st year were harvested and we call the stage Ratoon 1. After harvesting the sugarcane of Ratoon 1, the left root in the soil will grow up and the stems would be gathered, which we called Ratoon 2. In this experiment, we collected the data of characteristics for 3 years, including New Plant, Ratoon 1 and Ratoon 2, which are 3 replications of year.

1.2 Before the experiment, we put G×E interaction in consideration for the design. However, based on the following reasons, only one suitable plot was adopted.

1.2.1 As descriptions in the manuscript, we sampled a large number of materials, which needed to be treated by professionals and special equipment. Only Kaiyuan (the location of our institute) can meet our demands and the conditions of other places are limited.

1.2.2 Previous researches have documented that the main sugarcane producing areas in Yunnan, like Baoshao, Dehong, Lincang and Kaiyuan, possess the similar environmental and climatic characteristics. Furthermore, based on the research applied in the places mentioned above (Luo J., et al., 2015) ,G×E interaction showed no significance.

1.2.3 Our manuscript provides valuable information about the degradation of sugarcane varieties mainly caused by diseases, like sugarcane smut, mosaic disease, brown rust disease, spot disease and so on. Kaiyuan, as the main sugarcane breeding region in Yunnan, where the local climate is characterized as high humidity and temperature and this characteristics would contribute to the high rates of diseases. So, we think Kaiyuan could be served as the ideal research plot for our experiment.

Comments 2:

Furthermore, the author showed the parents of varieties but not show the inbreeding coefficient to evaluate genetic variation of the varieties. I think if the author show inbreeding coefficient should be better for the discussion.

Thank you for point this out. Here are our responses:

The genetic researches showed that the genetic background of modern sugarcane is similar, which is a general consensus. As discussed in our manuscript, foreign germplasm resources have played an important role in the cultivation of new sugarcane varieties in China, and according to incomplete statistics, from 1950 to 2000, 163 varieties were bred from 21 parents including CP, F and Co series, that accounting for 87.63% of the counted varieties and showing the high inbreeding coefficient. In addition, the research did in our research team showed that the genetic similarity coefficient among 98 varieties ranged from 0.73 to 0.75, and the highest value achieved to 0.93 (Zan et al., 2015). There are also many similar studies, which were cited in our manuscript, showing the higher inbreeding coefficient. Therefore, there is no associated results in the manuscript.

Reference

Luo J., Xu L.,Qiu J., Zhang H.,Yuan Z.,Deng Z.,Chen R.,Que Y. Evaluation of sugarcane test and ecological zone division in China based on HA-GGE Biplot(2015). Crop Journal, 41(02):214-227.

Zan F., Ying X., Wu., Zhao P., Chen X., Ma L., Su H., Liu J. Genetic diversity analysis of 98 collections of sugarcane germplasm with AFLP markers. Scientia Agricultura Sinica, 2015,48(05):1002-1010.

Reviewer 2 Report

The work has scientific merit for publication.
Adjustments are needed.
Some issues mainly in methodology need to be clarified.

 Some terms are overused in the text and need to be replaced.

Author Response

Response to Reviewer Comments

Comment 1

Some issues mainly in methodology need to be clarified.

Response 1:

We thank for your useful advice. Here are some major responses:

1.1 We added the information about how the sugarcane breeders were distributed in each characteristic and replication in line 134-135.

1.2 We have added the methodology applied with reference and the units for each characteristic in line 154-161.

1.3 Followed the suggestions of statistic part, we added more details in line 181-184.

1.4 The other detailed responses, please check Table 1.

Table 1. All responses to Reviewer 2

Comments and Suggestions for Authors

Line number

Authors' Reply

Modified line number

a question: these series are mentioned in Table 1?

84

21 parents including CP F and Co series were not mentioned in Table 1. Table 1 mainly included the information of test varieties and these test varieties was mainly of the 163 varieties were bred from 21 parents.

in the same year? December, 2016?

122

Added the world “next 3 years”

128

Indicate the year!

127

Had indicate the year in the document

132, 133

How the sugarcane breeders were distributed in the each trait and replication? This aspect in most important to reduced the interferences on the variation coefficient for statistical analysis!

128

Authors improved it according to the expert opinions and modifies it in the text. See the uploaded text for details.

134, 135

These grades were based on the anterior studies? Please, indicate the author (s) if possible!

132

Authors improved it according to the expert opinions and modifies it in the text. See the uploaded text for details.

139, 140

Some terms are overused in the text and need to be replaced.

142

“traits” substituted by “characteristics”

153

Indicate the methodology applied with reference and add the units for each characteristic

143

Authors improved it according to the expert opinions and modifies it in the text. See the uploaded text for details.

154-161

Add information about the test, and other statistical tool applied for example" the arcane process  then all data were normalized " mentioned at  Figure 1; Kruskal -Wallis...etc

165

Authors improved it according to the expert opinions and modifies it in the text. See the uploaded text for details.

181-184

include .... difference between the varieties for......

169

Authors improved it according to the expert opinions and modifies it in the text. See the uploaded text for details.

189-191

characteristics in 107 sugarcane varieties

173

Authors improved it according to the expert opinions and modifies it in the text. See the uploaded text for details.

204

group 2: ........, group 3: ......

184

Authors improved it according to the expert opinions and modifies it in the text. See the uploaded text for details.

214,215

cluster variety groups

203

Authors improved it according to the expert opinions and modifies it in the text. See the uploaded text for details.

252

characteristics between the four cluster variety groups

209

“traits” substituted by “characteristics”

261

substitute by significant role

229

Authors improved it according to the expert opinions and modifies it in the text. See the uploaded text for details.

282

substitute the "increase" term by other.....

253

substitute the "increase" term by “advance”

308

substitute the "increase" term by other.....

256

substitute the "increase" term by “advance”

311

substitute the "increase" term by other.....

257

substitute the "increase" term by “advance”

312

substitute the "increase" term by other.....

259

substitute the "increase" term by “advance”

314

substitute the "increase" term by other.....

260

substitute the "increase" term by “advance”

315

Comment 2

Some terms are overused in the text and need to be replaced.

The detailed responses please check Table 2.

Table 2 All responses to Reviewer 2

Comments and Suggestions for Authors

Line number

Authors' Reply

Modified line number

Some terms are overused in the text and need to be replaced.

8

Deleted “Correspondence:”

8

13

“traits” substituted by “characteristics”

13

21

“traits” substituted by “characteristics”

21

37

“traits” substituted by “characteristics”

37

66

“traits” substituted by “characteristics”

66

75

“traits” substituted by “characteristics”

75

122

“traits” substituted by “characteristics”

128

124

“traits” substituted by “characteristics”

130

131

“traits” substituted by “characteristics”

139

139

“traits” substituted by “characteristics”

150

168

“traits” substituted by “characteristics”

188

169

“traits” substituted by “characteristics”

189

Table 3

“traits” substituted by “characteristics”

Table 3

174

“traits” substituted by “characteristics”

205

176

“traits” substituted by “characteristics”

207

179

“traits” substituted by “characteristics”

210

Table 4

Add “%”. See the uploaded text for details.

Table 4

182

“traits” substituted by “characteristics”

213

188

“traits” substituted by “characteristics”

233

189

“traits” substituted by “characteristics”

234

191

“traits” substituted by “characteristics”

236

192

“traits” substituted by “characteristics”

237

201

“traits” substituted by “characteristics”

250

231

Deleted “played”

283

245

“revealed” substituted by “showed”

300

247

“traits” substituted by “characteristics”

302

249

“revealed” substituted by “indicated”

304

253

substitute the "increase" term by “advance”

308

255

“revealed” substituted by “showed”

310

256

substitute the "increase" term by “advance”

311

257

substitute the "increase" term by “advance”

312

259

substitute the "increase" term by “advance”

314

260

substitute the "increase" term by “advance”

315

266

“traits” substituted by “characteristics”

321

270

“traits” substituted by “characteristics”

325

312

“no” substituted by “not”

379

347

“alter” substituted by “change”

415

348

“in” substituted by “considering”

416

Round 2

Reviewer 1 Report

The answers of authors to my comments are acceptable.  However, if previous research (Luo J., et al., 2015) mentioned no G×E interaction among the main sugarcane producing areas in Yunnan, the author should note that results in the "materials and methods" or "discussion".
